# A Diagnostic Challenge in an Adolescent with Collagen VI-Related Myopathy and Emotional Disorder—Case Report

**DOI:** 10.3390/jpm13111577

**Published:** 2023-11-04

**Authors:** Mihaela Oros, Lucica Baranga, Adelina Glangher, Moldovan Adina-Diana, Gheorghita Jugulete, Carmen Pavelescu, Florin Mihaltan, Vasilica Plaiasu, Dan Cristian Gheorghe

**Affiliations:** 1Ponderas Academic Hospital, No. 85A, Nicolae G. Caramfil Street, 014142 Bucharest, Romania; mihaela.oros@prof.utm.ro (M.O.); barangalucica@yahoo.com (L.B.); adelina.glangher@gmail.com (A.G.); 2Physiology, Department of Preclinical Sciences, Faculty of Medicine, Titu Maiorescu University, No. 67A, Gheorghe Petrascu Street, 3rd District, 031593 Bucharest, Romania; rarinca.diana@gmail.com; 3Medlife SA, 365 Grivitei Bvd, 010719 Bucharest, Romania; 4Faculty of Medicine and Pharmacy, “Carol Davila”, No. 37, Dionisie Lupu Street, 2nd District, 020021 Bucharest, Romania; carmen.pavelescu@rez.umfcd.ro (C.P.); gheorghe.dancristian@gmail.com (D.C.G.); 5“Matei Balş” National Institute for Infectious Diseases, No. 1, Calistrat Grozovici Street, 2nd District, 021105 Bucharest, Romania; 6National Institute of Pneumology Marius Nasta, 050159 Bucharest, Romania; 7Regional Center of Medical Genetics, INSMC Alessandrescu-Rusescu, 020395 Bucharest, Romania; 8ENT Department “MS Curie” Hospital Bucharest, “Carol Davila” University of Medicine, 050474 Bucharest, Romania

**Keywords:** collagen VI-related disorders, pediatric sleep disorders, polysomnography, non-invasive ventilation, quality of life

## Abstract

Collagen VI-related disorders constitute a spectrum of severities from the milder Bethlem myopathy (BM) to the Ullrich congenital muscular dystrophy (UCMD), which is more severe, and an intermediate form characterized by muscle weakness that begins in infancy. Affected children are able to walk, although walking becomes increasingly difficult starting in early adulthood. They develop contractures in the ankles, elbows, knees, and spine in childhood. In some affected cases, the respiratory muscles are weakened, requiring mechanical ventilation, particularly during sleep. Individuals with collagen VI-related myopathy are at risk of restrictive lung disease and sleep-disordered breathing due to the development of scoliosis associated with neuromuscular weakness. Typical signs of respiratory failure are not always present, and some patients are unaware that their respiratory muscles have become weaker. Here, we report a case of an intermediate form of collagen VI-related myopathy confirmed by next-generation sequencing. The girl presented morning headache, irritability, and aggressiveness, and because of these main symptoms, she was referred by the neurologist for respiratory evaluation. The result of spirometry was associated with hypoventilation shown during sleep studies, indicating the necessity to initiate home non-invasive ventilation (NIV) with immediate improvement in the symptoms. Neuromuscular disorders (NMDs) have a great impact on sleep, but only very few studies evaluating sleep quality in young patients with collagen VI-related myopathy have been described. Daytime symptoms of sleep-disordered breathing may include irritability, emotional lability, and poor attentiveness, but these can be overseen by the severity of other complex medical problems in patients with collagen VI-related myopathy. We underline the importance of the close monitoring of respiratory function, sleep evaluation, and decision making to support the NIV treatment of other collagen VI-related myopathy variant-specific patients. Early recognition of sleep disturbances and initiation of respiratory support can preserve or enhance the quality of life for patients and their caregivers. Routine screening for identification of emotional distress should be instituted in the clinical practice using validated psychological measures in a multidisciplinary approach with different intervention strategies for both patient and parent when necessary.

## 1. Introduction

Collagen VI-related myopathy is a disease entity with a broad clinical spectrum that affects the skeletal muscles and connective tissue, caused by mutations in the α-chains of collagen-VI-related genes (*COL6A1*, *COL6A2*, and *COL6A3*) [1]. Collagen VI is a crucial part of the extracellular matrix, which, together with the surrounding basement membrane and cells, creates a microfibrillar structure. In many tissues, the extracellular matrix is essential for cell attachment, stability, and repair. Collagen VI-related disorders constitute a spectrum of severities from the milder Bethlem Myopathy (BM) to the Ullrich congenital muscular dystrophy (UCMD), which is more severe. Both BM and UCMD are now recognized as collagen VI-related myopathies, which have a wide range of clinical manifestations and were formerly thought to be distinct disease entities [1,2]. Given that the same gene mutation can cause a variety of clinical symptom severities, determining the genotype–phenotype association in this myopathy is particularly challenging. The diagnosis is made by using a combination of clinical presentation, muscle magnetic resonance imaging, muscle biopsy, and genetic analysis, which can be challenging to evaluate [3]. BM has a late onset and is often characterized by contractures of the ankle and long finger flexors, with modest phenotypic and a slow or stagnant progression. Hypotonia and postponed motor breakthroughs appear in early childhood [4]. More than two-thirds of those affected by BM remain freely mobile indoors while they are reliant on supportive means of transportation outdoors due to the sluggish onset of weakening. The intermediate form of collagen VI-related dystrophy is characterized by muscle weakness that begins in infancy (Table 1). They develop contractures in their fingers, elbows, shoulders, and ankles in childhood.

**Table 1 jpm-13-01577-t001:** The spectrum of clinical collagen VI-related conditions [5].

CONDITION	Bethlem Muscular Dystrophy (BMD)	Intermediate Form (Severe BMD/Mild UCMD)	Ullrich Congenital Muscular Dystrophy (UCMD)
Onset	Late onset	Late–early	Early onset, congenital
The main phenotype	Proximal muscle weakness and joint contractures	Not clearly defined	Congenital weakness, hypotonia, proximal joint contractures, and striking hyperlaxity of distal joints
Clinical picture	Mild hypotonia and weakness may be present congenitally. Hypotonia and delayed motor milestones occur in early childhood By adulthood, there is evidence of proximal weakness and contractures of the elbows, Achilles tendons, and long finger flexors	Individuals do not achieve the ability to run, jump, or climb stairs without the use of a railing The contractures are variable	Prenatal phenotype with decreased fetal movements was frequently reported
Prognosis	The progression of weakness is slow or static. More than 2/3 of affected individuals older than age 50 years remain independently ambulatory indoors while relying on supportive means for mobility outdoors	Independent ambulation past age 11 years	Progressive
Respiratory assessment	Respiratory involvement is not a consistent feature.	Respiratory insufficiency that is later in onset than in UCMD and results in the need for non-invasive ventilation in the form of bilevel positive airway pressure by the late teens to early 20s	Early and severe respiratory insufficiency occurs in all individuals, resulting in the need for non-invasive ventilation in the form of bilevel-positive airway pressure by age 11 years

The assessments listed in Table 2 are recommended to assess the severity of the disease and the needs of a person with collagen VI-related myopathy.

### Management of Clinical Manifestations in Collagen VI-Related Myopathies 

Scoliosis management and the treatment of Achilles tendon contractures are needed in individuals with collagen VI-related myopathy under the direction of an orthopedic specialist. Sometimes surgical treatment for scoliosis is required, and coordination between orthopedic, intensive care, and pulmonary specialists is necessary. Achilles tendon surgical procedures can increase the range of movements [3,4,6]. Physical therapy is recommended for joint stretches to improve the quality of life [5,7,8]. Neuromuscular disorders have a great impact on sleep. Sleep-disordered breathing is accurately detected by polysomnography (PSG), which also defines the severity of the condition. Muscle weakness with respiratory muscle dysfunction leads to low chest wall compliance with a decrease in lung volume, which causes hypoventilation [9,10,11]. Respiratory surveillance for nighttime hypoventilation is an important aspect of clinical therapy. Nocturnal hypercapnia can be predicted by forced vital capacity lower than 60% on spirometry [12]. During polysomnography (PSG), non-invasive ventilation (NIV) should be initiated using bilevel positive airway pressure (BiPAP), with pressures adjusted to provide adequate breathing, and then used constantly throughout the night. During recurrent PSG monitoring, it is important to carefully monitor and modify the nocturnal BiPAP pressures.

**Table 2 jpm-13-01577-t002:** Based on the severity of collagen VI-related myopathy and the system involved, the evaluation recommendation was summarized for the better management of patients [4,6,9,11,12].

System Involved	Evaluation	Recommendation
Muscle	Muscle weakness and how it affects movement and function, delayed motor milestones	Neuromuscular examination
Joints	Joint contractures are examined, especially to check for asymmetry in Achilles tendon contractures	Physiotherapy assessment
Skeletal	Rib cage deformities due to stiffness, scoliosis, respiratory muscle weakness	Orthopedic surgeon and pulmonary specialists’ evaluation
Pulmonary	Forced vital capacity must be constantly monitored, and sitting and supine positions will be used for pulmonary function tests PSG, CO_2_ monitoring	Pulmonary specialist and respiratory physiotherapy evaluation
Cardiovascular	Cardiologic abnormalities such as cardiomyopathy or symptomatic arrhythmia	Cardiac examination, echocardiogram, and ECG
Laboratory determinations	Genetic analysis	Consultation with clinical geneticist and genetic counselor
Skin	Cutaneous abnormalities such as keloids, keratosis pilaris, and soft/velvety skin	Specialist consultation

## 2. Methods 

We made a retrospective review of the medical record of a patient suffering from collagen VI-related myopathy. We measured body weight, height, arm span, body mass index (BMI), spirometry parameters (FVC-forced vital capacity and FEV1-forced expiratory volume in the first second as a percent predicted), and Cobb angle as indicators of growth status and respiratory function. Arm span was used as a stand-in for height for determining BMI and spirometry standards [9,13]. Spirometry data were collected using tools and procedures that adhered to the American Thoracic Society/European Respiratory Society’s norms Vitalograph 6800 Pneumotrac Spirometer with Spirotrac Software [13]. 

The nighttime video PSG was performed using System Alice 6 LDx, Phillips Respironics, with the software Sleepware G3 version 3.9.6.0. Transcutaneous CO_2_ (TcCO_2_) values were obtained over the entire PSG recording period of 485 min using the Radiometer Monitor system. 

Respiratory parameters, including apnea–hypopnea index (AHI) and sleep stages, were scored according to the American Academy of Sleep Medicine (AASM) criteria.

We collected the following respiratory parameters from the sleep study: total recording time (TRT), total sleep time (TST), apnea–hypopnea index (AHI), oxygen desaturation index (ODI), oxygen saturation nadir, and transcutaneous CO_2_ (TcCO_2_). This study was approved by the local ethical committee of the Regina Maria Healthcare Network Romania, No.372/31.05.2023, in accordance with the Declaration of Helsinki, and parents gave their written informed consent.

## 3. Case Presentation

We present a 12-year-old girl who was first referred to the sleep unit of the Pediatric Department in Ponderas Academic Hospital Bucharest for respiratory evaluation and sleep study using PSG, mainly for symptoms consisting of morning headache, irritability, and aggressive behavior.

From her history, we retained that she was born at term, from a normal pregnancy of a non-consanguineous healthy family. She presented with motor developmental delay, acquiring independent walking at the age of one year and a half with normal cognition. Starting with the second year of life, she showed chronic, progressive motor deficit involving all four limbs, clinically expressed as difficulties with both running and walking. 

Her current clinical exam showed weight = 43.5 kg (percentile 50), height = 159 cm (percentile 90) (arm span was used as a surrogate measurement for height), BMI = 17.2 kg/m^2^, thoracic dextroscoliosis with an asymmetric posture, with elevated right hip and bilateral high foot arches with hammertoes. The mental status was normal, although she appeared to be hesitating and somehow anxious; no psychiatric diagnosis was made. The clinical cardiovascular assessment was normal. The neurological phenotype consisted of muscle atrophy involving the upper limbs and the lower third of her thigh, with chronic, progressive muscle weakness, lower limbs being more affected (grade 3 on the Medical Research Council Scale for Muscle Strength scale) than upper limbs (grade 4, Medical Research Council Scale for Muscle Strength scale), with positive Gowers maneuver and joint retraction on both ankles (right side greater than left side) and shoulders. Cervical muscle weakness was observed, especially involving the neck flexors. Also, the patient presented with a particular gait with a tendency of asymmetric digitigrade walking, especially on the right limb, poor exercise tolerance, and could walk for distances no longer than 200 m, raising complaints of muscle fatigue and diffuse pain.

Molecular genetic testing was performed via sequence analysis and deletion/duplication testing of 71 gene panels associated with myopathies. A heterozygous variant in the *COL6A2 gene c.784 G > T*, *p.(Gly262Cys)* was detected, confirming the genetic diagnosis. It is classified as a likely pathogenic variant (class 2) according to the recommendation of the American College of Medical Genetics and Genomics guideline. This gene is related to an autosomal dominant and recessive Bethlem myopathy 1 (*BM; MIM #120250*) and Ullrich congenital muscular dystrophy 1 (*UCMD; MIM #158810)*. Although Sanger sequencing was not performed, her parents are clinically unaffected; thus, it is very probable to be the novo mutation.

A comprehensive assessment of the thorax and lungs was performed. The plain X-rays of the spine showed dextroscoliosis of the thoracic spine with 65 degrees Cobb angle and levoscoliosis of the lower thoracic and lumbar spine with 42 degrees Cobb angle along with axial dorsolumbar vertebral rotation (Figure 1). The sagittal alignment of the cervical spine was preserved. Also, a spirometry was performed 3 months before her previous admission, which showed altered respiratory parameters: FVC = 52.9% and FEV1 = 56.06% (in July 2019). At that moment, she was also evaluated via venous blood gas analysis: pH = 7.30, pCO_2_ = 56 mmHg. 

The results of the first PSG performed in September 2019 showed an apnea–hypopnea index AHI = 0.5/h, REMindex = 3.2/h, and maximum transcutaneous carbon dioxide pressure (PtcCO_2_) of 50.8 mmHg during sleep. Transcutaneous CO2 (TcCO2) values were obtained over the entire PSG recording period of 485 minutes using the Radiometer Monitor system, version 1.3.0. Details of the PSG parameters are given in Table 3. 

Low FVC was associated with increased respiratory effort with paradoxical breathing, low amplitude thoracic movements during sleep, and nocturnal hypercapnia, followed by daytime symptoms such as morning headaches, restless sleep, tiredness, and difficulty concentrating at school. All data were relevant for the diagnosis of sleep-related hypoventilation and justified the recommendation of NIV during sleep.

After 2 months, during her second visit to our department, a new PSG study was performed with BiPAP ST titration based on an acclimatization period with nasal mask Wisp X1 S/M. Using the manual titration protocol [14] and the patient’s compliance, expiratory positive airway pressure (EPAP)/inspiratory positive airway pressure (IPAP) was increased gradually to EPAP = 5 cmH20 and IPAP = 14 cmH20, back-up rate = 18 breathings/min, and Tinspiratory = 1 s (Table 3). 

A markedly favorable evolution was observed by the patient’s mother after the initiation of BPAP ST therapy during sleep, with the patient sleeping much better, with no aggressive behavior during daytime, with better school results, with no headaches, and improved tolerance to effort (which was visible during physical therapy).

The compliance card and therapy report showed that the patient wore the device 79.7% of the time in the first 5 months (138 days) after BiPAP therapy initiation, with an average of 6 h 36 min used each day and a maximum of 11 h 20 min (Table 4).

The next two PSG assessments demonstrated a satisfactory improvement both from a clinical perspective and by the resulting measurements (Table 3). The patient continues to be monitored multidisciplinary, including the evolution and the treatment of her scoliosis.

## 4. Discussion

Based on the clinical presentation and her evolution in time, our patient showed characteristics consistent with an intermediate form of collagen VI-related myopathy. 

In a recent study, Marinella et al. [15] proposed a diagnostic flowchart to assess the real pathogenicity of variants in collagen VI genes. The authors conclude that a multistep integration of clinical and molecular data allowed the identification of about 3% of those patients harboring pathogenetic collagen VI variants [15]. 

Nalini et al. prospectively evaluated nine cases of classical UCMD, primarily describing the motor milestones and other clinical phenotypic characteristics as histopathological and immunohistochemical findings [16]. They drew attention to the importance of prenatal diagnosis.

Highly developed research centers have contributed to the discovery of the connection between mitochondrial dysfunction [17,18] and defective autophagy [17,19] in the pathogenesis of collagen VI-related myopathies.

Merlini et al. conducted a cohort study in patients affected by collagen VI-related myopathies, providing clinical and genetic information and presenting quantitative data on muscle strength and contractures that can be valuable in clinical trials [17]. Their results show that respiratory failure can be a part of the clinical spectrum even in ambulatory patients, confirming the relationship between FVC and the three common collagen VI-related myopathies phenotypes acknowledged in previous studies [20,21]. They concluded that the age at which ventilation was initiated significantly differed among the three phenotypes.

Another study showed that respiratory surveillance is necessary for possible nocturnal hypoventilation, prophylaxis, and treatment of pulmonary infections [22].

The authors point out that assisted cough, non-invasive ventilation, and tracheostomy ventilation are treatments successfully applied in UCMD.

In a medical system that is ill-prepared for managing this kind of case and does not have a well-developed pediatric pulmonology network, the physician’s methodology is oriented toward early diagnosis. Therefore, in this case, by monitoring and improving the patient’s quality of life, an attempt was made to compensate for the major deficiencies of the medical system. Close contact with the patient, regular check-ups, and cultivating a trust-based relationship with the patient’s family were the priorities in this case, showing visible results upon its development.

NMDs have a great impact on sleep [23,24], but we found very few studies evaluating sleep quality in patients with collagen VI-related myopathy. They are especially related to the initiation of NIV in a patient who presented morning headache, irritability, and aggressiveness as main symptoms. 

Sleep-disordered breathing has been documented in children with neuromuscular weakness disorders that cause restrictive lung illness [9]. The severity of sleep-disordered breathing is dependent, in part, on lung volume, with a greater tendency for desaturation as lung volume is decreased [9].

Wang et al. mentioned in the Consensus Statement on Standard of Care for Congenital Muscular Dystrophies that the regular monitoring of pulmonary function can predict potential changes in a patient’s health [25]. The authors describe the indications and the utility of nocturnal oximetry, nocturnal CO_2_ monitoring, blood gas, and polysomnography including CO_2_ monitoring in the evaluation of respiratory function. 

Measurements of absolute lung volumes in older children with NMD consistently show significant decreases in vital capacity and functional residual capacity. Scoliosis is one of the most common and severe complications of virtually all conditions associated with muscle weakness, impacting respiratory mechanics in several ways [11,23] and additionally contributing to restrictive lung expansion, favoring the progression toward hypoventilation. Patients with scoliosis had lower minimal SpO_2_ values than the general population, and they experienced more apnea and hypopnea episodes during sleep than the control group [26]. 

The symptoms of nocturnal hypoventilation are divided into two categories [23,24,27]:Daytime symptoms: fatigue, excessive daytime sleepiness, morning headaches, fatigue upon awakening, reliance on alarms, frequent sleeping during weekends;Nighttime symptoms: snoring, apneas, restlessness, frequent awakenings, nocturia.

Our patient experienced agitated sleep, fatigue, difficult awakenings, dyspnea, morning headaches, concentration difficulties, decreased school performance, hypersomnolence, and altered mental status with normal SatO_2_ during wakefulness.

Sleep-disordered breathing symptoms due to muscle weakness have a slow progression and therefore present a high risk of being missed. In other words, it is possible that some patients do not realize they have lower respiratory muscle strength and that they may initially have subtle hypoventilation symptoms. Therefore, physical examination is not sufficient for the diagnosis of sleep-disordered breathing, and careful periodic monitoring is required, adapted to the progression of each case with collagen VI-related myopathy.

Polysomnography plays an important role in the diagnosis of sleep-disordered breathing in neuromuscular diseases and in optimizing non-invasive ventilation settings [23].

In our patient, we performed an initial PSG with TcCO_2_ and a systematic follow-up with PSGs after NIV initiation to evaluate the benefits of respiratory support for the progression of the underlying disease.

For monitoring gas exchanges during wakefulness and sleep, the non-invasive method of blood gas measurement is recommended. In more advanced stages, hypoxemia and CO_2_ retention may occur during the daytime. The earliest signs of hypoventilation occur during sleep, first in REM and then in NREM, with a subsequently progressive evolution, causing a decrease in Sat O_2_ and an increase in CO_2_ blood saturation [27,28,29]. 

Maximal PtcCO_2_ > 50 mmHg, with 3% oxygen desaturation index > 1.4 events/h and/or lung function data, has been used as a criterion to initiate NIV [28]. Besides these criteria for CPAP/NIV, others include the patient’s respiratory status and disease, abnormal daytime and nocturnal gas exchange, sleep and/or lung function data, or other parameters [29].

In our case with collagen VI-related myopathy, initiation criteria for BPAP therapy consisted of sleep hypoventilation symptoms, nocturnal hypercapnia, lung function with low FVC, and increased respiratory effort. We performed the titration PSG in order to determine the level of IPAP, EPAP, and backup rate. Therapeutic age-appropriate education played an important role and included both the patient and the caregiver.

NIV was associated with an improvement in sleep-related breathing disorder symptoms, nocturnal gas exchange, and improvement in the child’s and family’s quality of life. The patient’s compliance with therapy was good, especially after observing the positive effects of BPAP.

Darke et al. raised the issue that behavior, social, and communicational problems are common in children with neuromuscular diseases [30]. They draw attention to the fact that there is a higher likelihood that parents will seek advice about their children’s behavioral difficulties from the educational sector or at the muscle clinic than from mental health professionals.

Therefore, screening should be considered in order to identify children who are at high risk of emotional disorders and mental health problems. 

Limitations of this study are due to one case report information. In our study, pediatric pulmonologists played the role of facilitators of emotional health via the relationship based on trust between the patient and the family. Another limitation of our study is that we could not evaluate this patient using validated psychological measures. However, the patient is still being monitored and evaluated by a multidisciplinary team, so further data on her condition will be available in the future.

## 5. Conclusions

In patients with collagen VI-related myopathy, the caregivers and the medical team need to be aware of the possibility of unrecognized progressive respiratory problems. 

Daytime symptoms with morning headaches, tiredness, irritability, and difficulty concentrating at school can be relevant to identifying sleep-related hypoventilation and recommendations for NIV during sleep. 

According to the literature, the data integration of PSG results with clinical evaluation is important for the diagnosis and treatment of sleep-disordered breathing, with the final goal of preserving or improving the quality of life of patients and their caregivers. 

In our patient’s case, it has been demonstrated how the patient–medical doctor relationship can be beneficial for this patient’s future. This opens up the possibility for educational programs geared toward medical doctors, which emphasize early diagnosis and a proactive approach to the evolution of cases and promote emotional health care among patients with collagen VI-related myopathy and their families. 

Further studies will need to address mental health research. Routine screening for the identification of emotional distress should be instituted in the clinical practice in a multidisciplinary approach with different intervention strategies for both patient and parent when necessary.

## Figures and Tables

**Figure 1 jpm-13-01577-f001:**
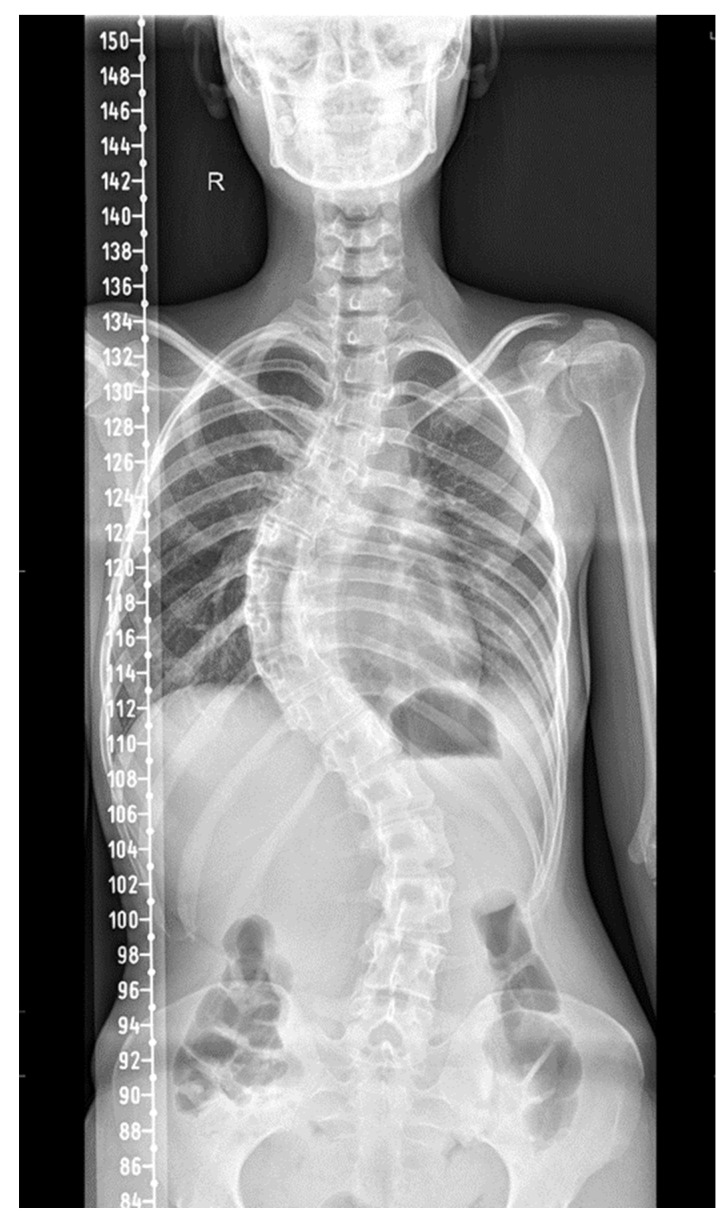
Thoracic dextroscoliosis and levoscoliosis of the lower thoracic and lumbar spine. (R = right side).

**Table 3 jpm-13-01577-t003:** Polysomnography findings.

Parameters PSG	T1	T2	T3
	1st BPAP Manual Titration with PSG (November 2019)	2nd BPAP (Re) Titration with PSG (7 Months from T1) (June 2020)	3rd BPAP (Re) Titration with PSG (8 Months from T2) (February 2021)
Total sleep time (min)	331	313.5	385.5
Sleep efficiency (%)	68.8	84.9	87.9
Stage N1 (%)	6.5	4.9	4.9
Stage N2 (%)	41.2	55.8	63.2
Stage N3 (%)	38.4	25.4	24pr.3
Stage R (%)	13.9	13.9	7.7
AI (events/h)	4.7	9.2	14
3% ODI (events/h)	2.5	0.6	0.8
AHI index (events/h)	0.9	0.4	0
REMindex (events/h)	0.0	1.4	0
Wake SpO_2_ (%)	97	97	97
Mean SpO_2_ (%) sleep	96	97	96
Nadir SpO_2_ (%) sleep	92	94	94
Max PtcCO_2_ in sleep (mmHg)	50.9	45.3	48.4
Sleep respiratory rate	15–20	16–20	18–19
Settings at the end of the titration	IPAP = 14 cm H_2_O EPAP = 5 cm H_2_O Back up rate = 18 r/min Tins = 1 s	IPAP = 15 cm H_2_O EPAP = 5 cm H_2_O Backup rate = 18 r/min Tins = 1 s	IPAP = 16 cm H_2_O EPAP = 5 cm H_2_O Backup rate = 18 resp/min Tins = 1 s
	Nasal mask	Nasal mask	Nasal mask

Abbreviations: PSG—polysomnography, oAHI—obstructive apnea-hypopnea index, ODI—oxygen desaturation Index, AI—arousal index, REM—rapid eye movement, SpO_2_—oxygen saturation, PtcCO_2_—transcutaneous carbon dioxide pressure, IPAP—inspiratory positive airway pressure, EPAP—expiratory positive airway pressure, Tins—inspiratory time.

**Table 4 jpm-13-01577-t004:** The compliance card and therapy report in the first 5 months after BiPAP therapy was initiated.

Compliance Summary	
Date Range	1 February 2020–17 June 2020 (138 days)
Days with Device Usage	110 days
Days without Device Usage	28 days
Percent Days with Device Usage	79.7%
Cumulative Usage	30 days 7 h 5 min 52 s
Maximum Usage (1 Day)	11 h 20 min 1 s
Average Usage (All Days)	5 h 16 min 7 s
Average Usage (Days Used)	6 h 36 min, 35 s
Minimum Usage (1 Day)	2 min 33 s
Percent of Days with Usage ≥ 4 h	73.2%
Percent of Days with Usage < 4 h	26.8%
Total Blower Time	30 days 7 h 20 min 7 s

## Data Availability

All data supporting this article are publicly available and can be retrieved from repositories and platforms such as PubMed.

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
