# Peer review of "A Diagnostic Challenge in an Adolescent with Collagen VI-Related Myopathy and Emotional Disorder—Case Report"

_jpm, 2023, doi:10.3390/jpm13111577_

Round 1
Reviewer 1 Report
Comments and Suggestions for Authors
The manuscript includes a detailed description of the collagen VI-related myopathy at a medical-biological level, intervention performed, etc. in a teenage patient, although the need for psychological-functional exploration could be further explored. These questions would be fundamental to offer a broad approach to the factors that influence the health of these patients.
As an example, considering the patient's clinical condition, the need to examine the emotional manifestations (which can be intense, even if there is no psychiatric diagnosis) could be indicated since they can have an impact on the quality of life of those affected by this pathology.
It would be interesting and useful for readers of published clinical cases to make a practical description of how to carry out a global assessment of different factors that can affect the health status of these patients (e.g., raise the need for psychometric evaluation of the intensity of anxiety-depressive symptoms, relevant screening tests for adolescents, specific assessments of quality of life, exploration of sleep quality using standardized tools, etc.) in addition to the physical impact associated with the pathology.
The limitations inherent to the descriptive study of a single case should be delved into (since it is only indicated). The need to explore other conditions to a greater extent (considering what was previously indicated) could also be considered, along with social functioning, contextual impact of the pathology, etc. since it is indicated that it is being evaluated by a multidisciplinary team that will allow us to know to a greater extent the impact and development of the pathology.
Lastly, the table 4 should be presented in the same format in which tables 1, 2 and 3 have been presented, since it appears as a figure. On the other hand, the patient consent form should be reviewed to verify that it is complete in all the requested sections, since it is a relevant aspect at a procedural level in this type of publications.
Author Response
Thank you for your time and recommendations, we hope to answer to all the requests,
best regards,

Reviewer 2 Report
Comments and Suggestions for Authors
Manuscript Title : A diagnostic challenge in an adolescent with collagen VI-related myopathy and emotional disorder - case report
Manuscript ID : jpm-2641468.
The reviewed paper reported the case of a young patient presenting an intermediate form of collagen VI-related myopathy. The study focused on close monitoring of respiratory function, sleep evaluation, and decision-making to support the NIV treatment of collagen VI-related myopathy.
The topic discussed in this paper is very interesting and the diagnosis work is important. The language of the manuscript is acceptable. Results found in this research work are of great interest.
The paper could be considered for publication in Journal of Personalized Medicine after authors address the below given points:
* Explain clearly the novelty of the present work.
* Compare obtained results with other similar case(s).
* Detailed comments are reported on the attached file : Review.jpm-2641468.pdf

Author Response
We hope that we managed to bring those revisions as you expected.thank you,

Reviewer 3 Report
Comments and Suggestions for Authors
The problem of early diagnosis of collagen VI-related myopathies is relevant from a practical point of view. I think the case report will be useful for clinicians. However, the manuscript needs a serious stylistic and technical revision. In the present form, the manuscript is difficult to read, it is desirable to structure the content.
Introduction
Line 53: Write the name of the genes in italics here and further.
Lines 57, 63, 87, 88, etc.: When using abbreviations for the first time, explain them. Please do not use abbreviations if they are used 4 or less times in your manuscript. Check all abbreviations twice.
Line 72: In Table 1, use the full name OR abbreviation in the column names and in the table cells. If you decide to use abbreviations, then add a Note under the table where you explain all the abbreviations that were used in this table. Please use the MDPI template, remove all bold text highlighting except the column names. The contents of the table need a technical and stylistic revision.
Add the purpose of your clinical report.
Results
Lines 118, 127, etc.: You have previously explained these abbreviations to PSG and IBM. It is not necessary to explain abbreviations twice. Check the use of all abbreviations in the text of the manuscript twice.
Lines 139-145: Is the pathogenicity of the detected mutation confirmed in the patient? Whether Sanger sequencing was performed (trio method: the patient, her father and mother)? Are the patient's parents healthy? Are they carriers of a causal mutation? Or is this mutation de novo?
Lines 171 – 173: Move the explanation of all abbreviations from the name of Table 2 to the Note under this table.
Lines 182 -184: Table 4 needs a technical revision. Use the MDPI template. Add the name of the second column.
Line 141 – Write the name of the gene in italics.
Discussion
How does this clinical case differ from the previously published ones? What is its uniqueness? What were the main problems of diagnosing the disease in this patient?
References
13 links need to be updated because they were published more than 10 years ago. Links of historical interest can be left, but their number needs to be reduced.
Comments on the Quality of English LanguageThe style of the English language needs revision.
Author Response
thank you for your review, regarding your recommendations, we hope everything will be as you requested. if any other additions are needed, we are here.

Round 2
Reviewer 1 Report
Comments and Suggestions for Authors
Thank you for considering and including the requested modifications, the manuscript is much more appropriate in the revised version.
Reviewer 2 Report
Comments and Suggestions for Authors
The authors addressed well the requested issues according to our guidelines. The language of the manuscript was obviously improved. So, the paper could be published in JPM in its present form.